# Targeting mosquito X-chromosomes reveals complex transmission dynamics of sex ratio distorting gene drives

Daniella An Haber [1,7], Yael Arien[1,7], Lee Benjamin Lamdan[1], Yehonathan Alcalay[1], Chen Zecharia[1], Flavia Krsticevic[1], Elad Shmuel Yonah[1], Rotem Daniel Avraham[1], Elzbieta Krzywinska[2,3], Jaroslaw Krzywinski [2,4], Eric Marois [5], Nikolai Windbichler [6] & Philippos Aris Papathanos [1] ✉

Engineered sex ratio distorters (SRDs) have been proposed as a powerful component of genetic control strategies designed to suppress harmful insect pests. Two types of CRISPR-based SRD mechanisms have been proposed: X-shredding, which eliminates X-bearing sperm, and X-poisoning, which eliminates females inheriting disrupted X-chromosomes. These differences can have a profound impact on the population dynamics of SRDs when linked to the Y-chromosome: an X-shredder is invasive, constituting a classical meiotic Y-drive, whereas X-poisoning is self-limiting, unable to invade but also insulated from selection. Here, we establish X-poisoning strains in the malaria vector *Anopheles gambiae* targeting three X-linked genes during spermatogenesis, resulting in male bias. We find that sex distortion is primarily driven by a loss of X-bearing sperm, with limited evidence for postzygotic lethality of female progeny. By leveraging a *Drosophila melanogaster* model, we show unambiguously that engineered SRD traits can operate differently in these two insects. Unlike X-shredding, X-poisoning could theoretically operate at early stages of spermatogenesis. We therefore explore premeiotic Cas9 expression to target the mosquito X-chromosome. We find that, by pre-empting the onset of meiotic sex chromosome inactivation, this approach may enable the development of Y-linked SRDs if mutagenesis of spermatogenesis-essential genes is functionally balanced.

In 1967, Hamilton proposed that sex ratio distorting (SRD) alleles could represent a component of a highly efficient genetic control method for the suppression of pest species, highlighting malaria mosquitoes as a possible target[1]. Approximately forty years later, insect molecular biology and genetics had matured sufficiently to begin engineering Hamilton's meiotic drives in the malaria vector *Anopheles gambiae*.

Using the I-*Ppo*I meganuclease from a slime mold, we targeted the repetitive and X-chromosome-specific 28S rDNA gene cluster during male meiosis, a strategy we called X-shredding[2,3]. As a result, X-bearing gametes were almost entirely excluded from successfully fertilized eggs, resulting in a highly male-biased sex ratio within the offspring of transgenic males. Subsequently, CRISPR/Cas9 was used instead of

[1]Department of Entomology, Institute of Environmental Sciences, Robert H. Smith Faculty of Agriculture, Food and Environment, Hebrew University of Jerusalem, Rehovot 7610001, Israel. [2]The Pirbright Institute, Ash Road, Pirbright, Surrey GU24 0NF, UK. [3]Forest Research, Alice Holt Lodge, Farnham, Surrey GU10 4LH, UK. [4]Genetics and Ecology Research Centre, Polo d'Innovazione di Genomica Genetica e Biologia, Via Mazzieri, 05100 Terni, Italy. [5]Institut de Biologie Moléculaire et Cellulaire, Université de Strasbourg, INSERM, CNRS, Strasbourg, France. [6]Department of Life Sciences, Imperial College London, London SW7 2AZ, UK. [7]These authors contributed equally: Daniella An Haber, Yael Arien. ✉e-mail: p.papathanos@mail.huji.ac.il

I-*Ppo*I, targeting the same 28S rDNA genes, and also resulted in high sex ratio distortion without affecting male fertility[4]. This opened new possibilities to transfer this system to other pests, which do not share the unique X-chromosome specificity of the rDNA cluster of *An. gambiae*, by targeting other repetitive X-chromosome-specific sequences. To test this, we previously identified[5] and then targeted suitable repetitive X-linked sequences first in the *Drosophila melanogaster* model[6], and soon after in the agricultural pest *Ceratitis capitata*[7], thereby confirming that CRISPR-based SRDs can be transferred to additional important pests. Insertion of an active X-shredding SRD into an insect Y-chromosome in the following step would result in its shielding from negative selection in females and ensure transmission to all male progeny, realizing the population suppression meiotic drive originally described by Hamilton in his models.

However, because a highly invasive Y-linked X-shredder may not always be desirable, for example, by spreading to non-target areas, or simply because gene drives are controversial due to their novelty and implications, Burt and Deredec[8] proposed an alternative Y-linked SRD design based on the targeting of X-chromosome genes with a haploinsufficient (HI) phenotype. Since X-chromosomes are exclusively transmitted to female progeny from males with XY karyotypes, developmental lethality would be confined to females, resulting in their postzygotic mortality. Alternatively, autosomal HI genes could be targeted, provided they are female-specific. Similarly to X-shredding, Y-linkage would also shield such a postzygotic SRD from negative selection in females and ensure transmission to all males, but unlike X-shredding, the lethality of all female progeny would theoretically limit the invasiveness of this system. More specifically, when the dynamics of suppression and transmission of such a Y-linked SRD are modeled, its behavior reflects a self-limiting system that does not spread but which can persist at its initial release frequency. This persistence is unlike other self-limiting constructs, such as female-specific RIDL, which decline in frequency and ultimately disappear without additional releases. Given these interesting dynamics, an autosomal proof-of-concept for such a system was demonstrated in *D. melanogaster*[6]. To distinguish it from X-shredding we called this approach X-poisoning, reflecting the postzygotic lethality of females inheriting mutated X-chromosomes from their fathers (Fig. 1). X-poisoning transgenic males were designed to express Cas9 in sperm and sgRNAs targeting X-linked ribosomal protein (RP) genes with a predicted HI phenotype. Postzygotic lethality of a significant fraction of the offspring from X-poisoning males was observed and corresponded with a highly male-biased sex ratio among survivors reaching adulthood (92% males), indicative of developmental lethality of females receiving mutant X-linked alleles. In a second *Drosophila* study, targeting a different X-linked HI gene involved in wing development also resulted in significant postzygotic lethality of female offspring[9]. Here, we develop and test the X-poisoning system in the malaria vector *An. gambiae* using proof-of-concept autosomal strains. We discover significant shifts of normal sex ratios, but through unexpected mechanisms that modify transmission dynamics of sex chromosomes. Using our *Drosophila* system we perform direct comparisons between X-poisoning in mosquitoes and flies and confirm that Cas9-targeted X-chromosomes are transmitted differently to the next generation between these two related insect species. Based on these results we demonstrate how this unexpected mechanism can help to close a current technological barrier for the engineering of invasive Y-chromosome drives in mosquitoes.

## Results

### Generation of transgenic mosquito X-poisoning strains

Cytosolic ribosomal protein (RP) genes have known haploinsufficient (HI) mutant phenotypes in many species, including in *Saccharomyces* yeast and *D. melanogaster,* and were the basis of our first X-poisoning designs in flies[6,10]. *An. gambiae* and *D. melanogaster* do not share an ancestral dipteran X-chromosome, with the ancestral dipteran X being the dot chromosome in *D. melanogaster*[11,12]. Hence, we did not assume that we could directly target mosquito RP orthologs of validated *Drosophila* X-poisoning RPs, as these would likely not be X-linked in mosquitoes. We therefore cataloged all mosquito RP genes, identifying six candidate loci on the X-chromosome (Fig. 2a). Among the six candidates, we designed X-poisoning sgRNAs to target AGAP000739 (*AgRpS10*), AGAP000952 (*AgRpL37*) and AGAP000953 (*AgRpL10-1*) (Fig. 2b). We prioritized AGAP000952 and AGAP000739 because both have sperm-specific, autosomal paralogs (AGAP011501 and AGAP005469, respectively) (Figure S1a) that may have evolved to substitute the function of the X-linked RP genes in meiotic cells, as has been observed in other species[13,14]. We reasoned that autosomal paralogs may be able to functionally rescue the targeting of the X-linked gene by Cas9, if the targeted X-linked gene is essential for spermatogenesis. This could mitigate trade-offs between Cas9 activity and male fertility and enable the X-poisoning effect to manifest only in the fertilized embryo. We also compared the relative expression of the non-targeted autosomal paralogs versus the targeted X-linked RP genes throughout stages of sperm development (Figure S1b). AGAP000953 was selected for its proximity to AGAP000952 and because its autosomal paralog (AGAP002395) is identical in protein sequence to the X-linked gene (Figure S2), suggesting the autosomal paralog might be able to completely rescue the function of the X-linked gene. While AGAP002395 is not sperm-specific due to its expression in other tissues (Figure S1a), it is strongly expressed in the sperm throughout spermatogenesis (Figure S1b). *piggyBac* transformation constructs were designed to express either one or two sgRNAs per X-linked target using a previously described promoter of a ubiquitously expressed *U6* gene[15]. Cas9 expression was driven using the regulatory regions of the spermatogenesis-specific *β2-tubulin* gene, to restrict Cas9 activity to spermatogenesis similarly to previous X-shredding constructs[2–4,6,16]. In addition, each construct contained a 3xP3-DsRed transformation marker[17]. For the single sgRNA constructs, we were able to isolate three independent transgenic strains targeting AGAP000739 (739_1_a, 739_1_b, and 739_1_c), two targeting AGAP000952 (952_2_a and 952_2_b), and one targeting AGAP000953 (953_1_a). For the double sgRNA construct, we generated a construct combining the two top-ranking sgRNAs for AGAP000952, based on predicted scores, resulting in independent strains 952_1;952_2_a, and 952_1;952_2_b. Finally, to test the combined effects of X-poisoning and X-shredding, we generated a strain expressing an sgRNA targeting AGAP000952 along with the previously validated X-shredding sgRNA targeting the 28S rDNA[4] (strain 952_1;Xshrd). In total, we generated nine independent transgenic strains harboring multiple designs (Fig. 2c and e).

### Evaluating the sex ratio in offspring of X-targeting strains

To test the performance of our X-targeting strains, transgenic heterozygous males were crossed to wild-type females and offspring were reared to adulthood to measure the sex ratio. Reciprocal crosses of transgenic heterozygous females and wild-type intercrosses served as controls (Fig. 2d). Of the nine strains, three exhibited a significant male-bias among offspring reaching adulthood compared to both the wild-type cross and their reciprocal female cross (Tukey-Kramer, $p < 0.0001$) (Fig. 2e). The first strain, 739_1_c, corresponds to a single sgRNA construct targeting AGAP000739. Among the three independent strains expressing the same sgRNA (739_1), only 739_1_c demonstrated a significant male bias in the F1 populations (67.2%). The second strain, 952_1;952_2, corresponds to the double sgRNA construct targeting AGAP000952. Among the two independent strains expressing the same sgRNAs, only 952_1;952_2_a exhibited substantial sex ratio distortion (73%). The distortion levels observed in the two X-poisoning strains, 739_1_c and 952_1;952_2_a, were not statistically different (Tukey-Kramer, $p = 0.2652$). The third strain, 952_1;Xshrd,

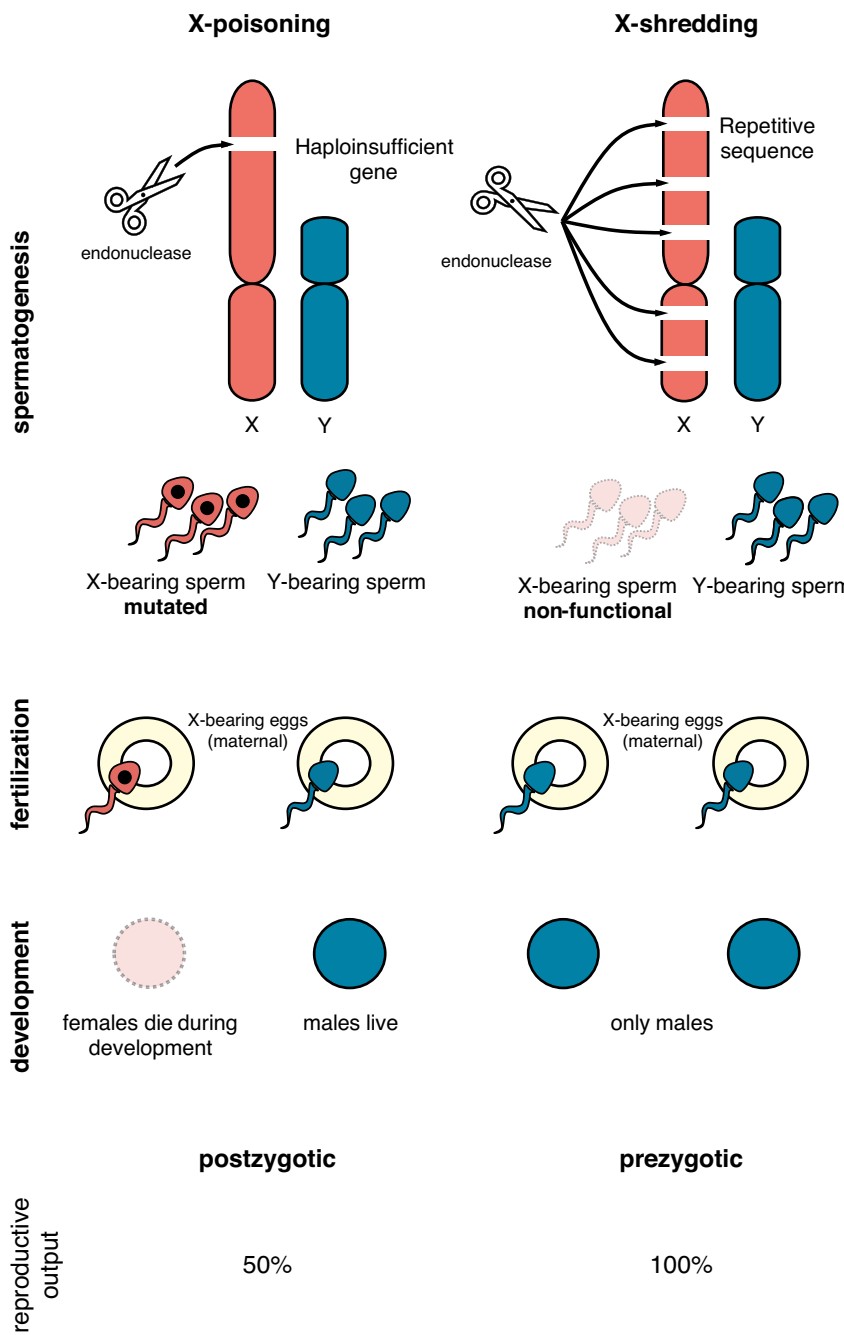

**Fig. 1 | Mechanisms of synthetic CRISPR-based sex ratio distorters (SRDs).** A sex ratio distorting CRISPR endonuclease targeting the X-chromosome during spermatogenesis could act in one of two modes: (1) by mutating X-linked essential haploinsufficient genes, resulting in postzygotic lethality of females inheriting targeted chromosomes (X-poisoning, left) or (2) by repeatedly cutting the X-chromosome, leading to a prezygotic meiotic drive through the loss of X-bearing sperm (X-shredding, right). Blue—Y-chromosome, Y-chromosome-bearing sperm, XY (male) offspring; Red—X-chromosome, X-chromosome-bearing sperm, XX (female) offspring; Transparent red with dashed outline—nonfunctional sperm or non-viable offspring; Yellow—X-bearing egg. A black dot indicates sperm carrying modifications at the X-linked target gene.

corresponds to a double sgRNA construct combining both X-poisoning and X-shredding. With an average of 80.8% male F1 adults, the sex ratio distortion in 952_1;Xshrd was statistically different from both 739_1_c (Tukey-Kramer, $p < 0.0001$) and 952_1;952_2_a (Tukey-Kramer, $p = 0.0154$). Next, we dissected testes from transgenic male adults and ran total mRNA sequencing to quantify Cas9 expression levels and to relate these to distortion phenotypes (Figure S3a). Cas9 expression levels in testes were generally predictive of sex ratio distortion, where strains of the same construct resulting in significant distortion levels consistently had higher Cas9 expression compared to those that did not display sex ratio distortion. For strains displaying male bias, we also performed amplicon sequencing from pools of dissected testes and surviving female progeny (Figure S3b and Figure S4). Amplicon sequencing for males of all living transgenic strains, regardless of the distortion phenotype, was also performed using genomic DNA extracted from posterior adult male abdomens that contain testes, but also other tissues, instead of dissected testes, given the larger number of strains (8 of 9). The frequencies of unique alleles not perfectly matching the native sgRNA target sites and occurring at least 10 times in the data were quantified. Mutant alleles were found at all sgRNA target sites, suggesting that all selected sgRNAs can guide Cas9 activity against their respective target sites (Figure S3b). Mutant

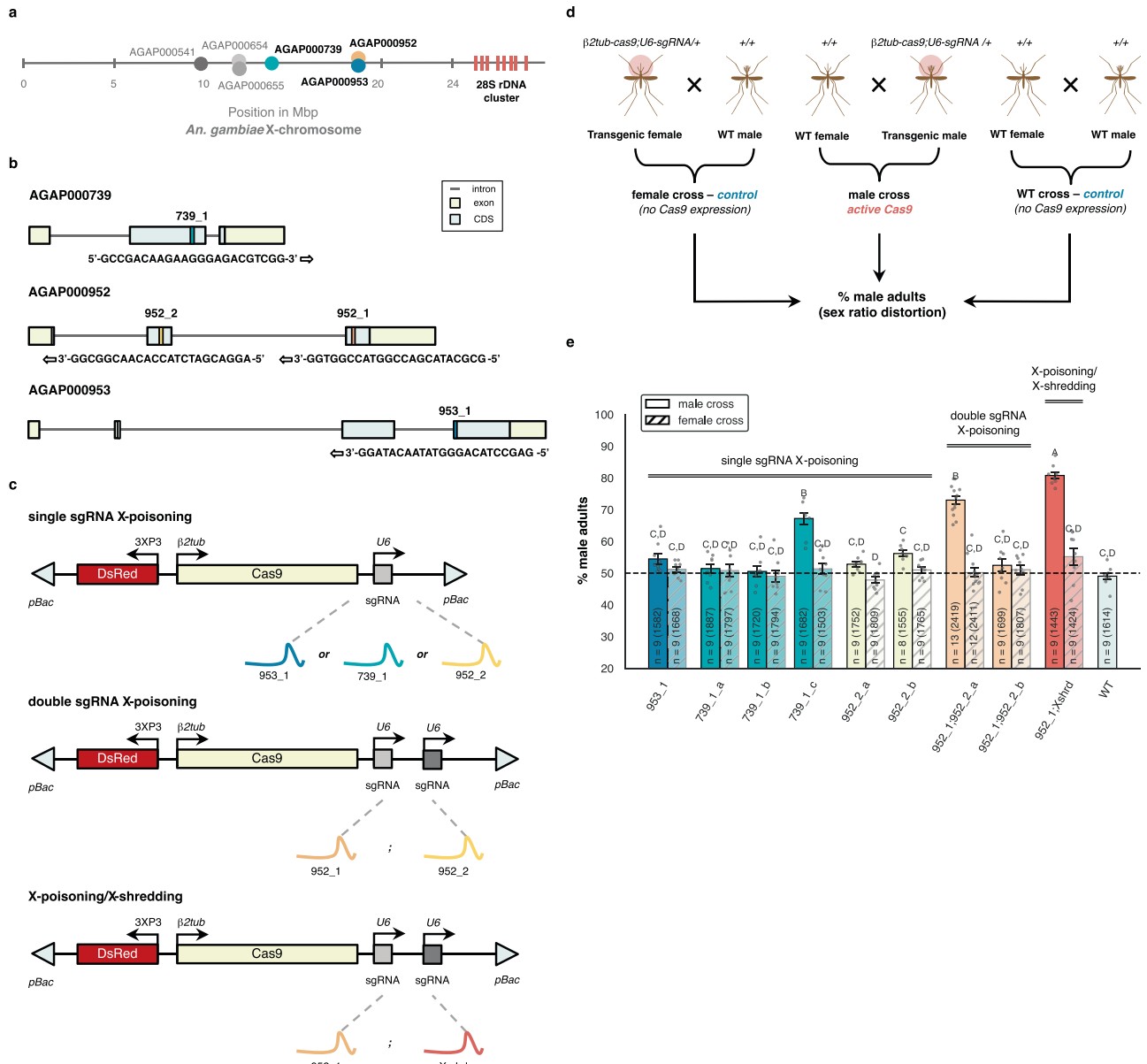

**Fig. 2 | Development of X-chromosome targeting SRD strains in *An. gambiae*.**
**a** Relative position of candidate haploinsufficient target genes and the 28S rDNA cluster on the *An. gambiae* X-chromosome. **b** Structure of targeted X-linked ribosomal protein genes indicating exons, introns, and coding sequences (CDSs). The location, direction, and sequence of sgRNA target sites are also presented, with PAM sites shown underscored. **c** Schematic representation of transformation constructs. A total of five distinct constructs were generated, from three different types (single sgRNA X-poisoning, double sgRNA X-poisoning, and X-poisoning/X-shredding). (*pBac*) piggyBac inverted repeats; (3xP3-DsRed) the 3xP3 promoter driving DsRed fluorescent marker; (*β2tub*-Cas9) codon-optimized CDS of Cas9 endonuclease under the control of the male germline specific *β2tubulin* promoter; (*U6*-sgRNA) sgRNA under the control of the ubiquitous *U6* Pol III promoter. **d** Overview of experimental crosses. Transgenic mosquitoes are indicated with a semi-transparent red circle, indicating 3xP3-DsRed expression. (WT) wild-type

**e** Percentage of F1 adult males within each independent transgenic strain and the wild-type strain (WT). Data are presented as mean values +/-SEM. A total of 9 independent transgenic strains were generated (X-axis), with strains carrying the same transgenic constructs being colored the same. For every transgenic X-targeting strain, both male (solid color) and female (transparent with hatch) crosses are shown. "n" indicates the number of biological replicates used to derive statistics, with the total number of adult F1 individuals that were counted and sexed per cross provided in brackets. Statistical significance was derived from a One-Way ANOVA of the percentage of F1 adult males by strain/cross combination (F(18, 158) = 37.6, *p* < 0.0001), followed by Tukey-Kramer HSD for comparisons between all pairs (α = 0.05). Different letters indicate statistical difference using Tukey-Kramer HSD. Individual data points represent data from each replicate. Source data are provided as a Source Data file.

alleles were consistently more abundant in transgenic testes or terminal segment samples than those occurring in wild-type samples and were composed of substitutions and both in- and out-of-frame edits. When mutant alleles were identified in surviving females, they were exclusively composed of substitutions and in-frame deletions, suggesting that null or deleterious alleles of the X-linked target genes produce HI phenotypes, as expected, and that mutated

X-chromosomes are transmitted to females from transgenic males (Figure S4).

**Evaluating the timing of female elimination by X-poisoning**
According to our model and data in *Drosophila* (Fig. 1), X-poisoning should result in female-specific lethality during development, as an outcome of Cas9-induced mutagenesis of X-linked RP genes that are

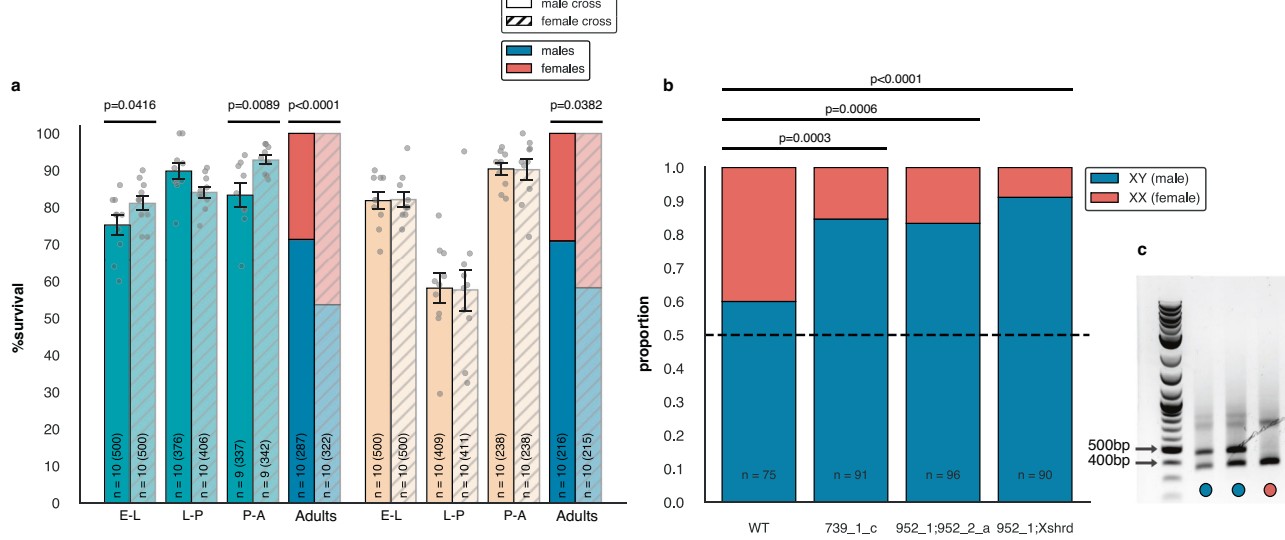

**Fig. 3 | Sex ratio distortion and the timing of female elimination. a** Egg-adult survival analysis of F1 populations from active X-poisoning strains. The mean survival rate (+/-SEM) at each developmental stage and the mean percentage of male adults are shown for F1 populations from both male (solid color) and female (transparent with hatch) crosses for each strain. E-L—Egg to Larva transition (hatching); L-P—Larva to Pupa transition (pupation); P-A—Pupa to Adult transition (adult emergence). "n" indicates the number of replicates used to derive statistics, with the total number of eggs, hatching larvae, pupae, and adults counted at each stage, respectively, provided in brackets. Statistical significance was derived from one-sided t-tests, assuming equal or unequal variances, comparing the survival rates at each stage and the percentage of F1 adults between the male and female crosses ($\alpha = 0.05$). Individual data points represent data from each replicate. p-values are indicated within the figure. **b** PCR-based genotyping of individual *An. gambiae* eggs using sex-specific markers. Data are shown as the proportion of XY (male) and XX (female) eggs in F1 populations of the wild-type cross (WT) and of male crosses from active SRD strains. "n" represents the total number of eggs that were successfully genotyped in each cross. Statistical analysis was done using a one-sided Fisher's Exact Test to compare the proportion of XY (male) eggs in each transgenic cross to the wild type control ($\alpha = 0.05$). p-values are indicated within the figure. **c** Examples of XY (male) and XX (female) PCR patterns used for genotyping analysis. Source data are provided as Source Data files.

transmitted exclusively to females from transgenic males. To evaluate whether, and at what developmental stage, female-specific lethality occurs in our mosquito strains, we designed an experiment to track offspring survival throughout development in male crosses of the two active X-poisoning strains compared with their reciprocal female crosses. Surprisingly, we did not observe a decrease in relative offspring survival between male and female crosses that could account for final adult sex ratios (Fig. 3a). More specifically, when comparing the male cross of the X-poisoning strain 739_1_c with the reciprocal female cross, we observed only a slight reduction in hatching and adult emergence (t-test, 6% with $p = 0.0416$ and 9.6% with $p = 0.0089$, respectively). Strikingly, in offspring from male crosses of the 952_1;952_2_a strain there was no difference in F1 survival between male and female crosses at any stage of development (Fig. 3a). The sex ratios at the adult stage in stains 739_1_c and 952_1;952_2_a were consistent with our previous findings, exhibiting male bias in male crosses compared to female crosses (t-test, $p < 0.0001$ and p = 0.0382, respectively) (Fig. 3a). We observed relatively low survival of larvae that became pupae in offspring of 952_1;952_2_a parents, but concluded that it was independent of SRD activity, as it manifested in both the male and female reciprocal control cross and not in subsequent experiments (see below).

The absence of obvious postzygotic lethality during development coupled with the effective sex ratio distortion led us to consider the possibility that targeting loci on the X-chromosomes, even at single site, could result in meiotic drive through the removal of X-bearing gametes similarly to X-shredding, favoring the transmission of Y-bearing sperm (Fig. 1). To investigate this, we genotyped individual embryos from crosses between SRD males and wild type females using diagnostic PCR primers that bind to abundant sex chromosome-linked satellite DNAs AgX367 and AgY477[18]. Using these primers, amplification from the Y-chromosome yields a male-specific 477 bp product, whereas

amplification from the X-chromosome produces a non-sex-specific 367 bp product, allowing us to easily discriminate between male (XY) and female (XX) genomic DNA (Fig. 3c). For both X-poisoning strains, the proportion of male embryos out of the total number of embryos tested was significantly greater than in the wild type cross (Fisher's Exact test, $p = 0.0003$ and $p = 0.0006$ for 739_1_c and 952_1;952_2_a, respectively), suggesting that biased transmission of Y-bearing sperm may be the mechanism of sex ratio distortion, resulting in the absence of clear postzygotic lethality (Fig. 3b). We also performed this analysis with the combined X-poisoning/X-shredding strain 952_1;Xshrd and observed similar results (Fisher's Exact Test, $p < 0.0001$) (Fig. 3b).

## Tracking Y-chromosome transmission and sex-specific survival in mosquito and fly SRD strains

Given these unexpected results, we sought to examine further how Cas9 targeting of mosquito X-linked genes affects sex chromosome transmission dynamics, and to track in more detail the sex-specific survival of offspring. More specifically, we were interested in precisely quantifying sex-specific survival of male and female progeny from each of the male crosses, rather than measuring overall survival rates agnostic of offspring sex, to see if the sex ratio differences are manifested in newly-hatched larvae and to see if hatching rates and sex ratios are coupled. To discriminate between male and female individuals from first-instar larval stage (L1), we crossed males from a transgenic strain carrying a Y-linked eGFP marker (Ygfp) to females of the 739_1_c, 952_1;952_2_a, and 952_1;Xshrd strains, to obtain males carrying both a Y-chromosome linked marker and the CRISPR SRDs (Fig. 4a). We then established male and control (Ygfp-only males) crosses and tracked the percentage of male L1 larvae, hatching rates, and the lethality of both sexes throughout development. A significantly higher frequency of males (eGFP + ) was detected among newly hatched L1 larvae with all strains compared to control (Dunnett's

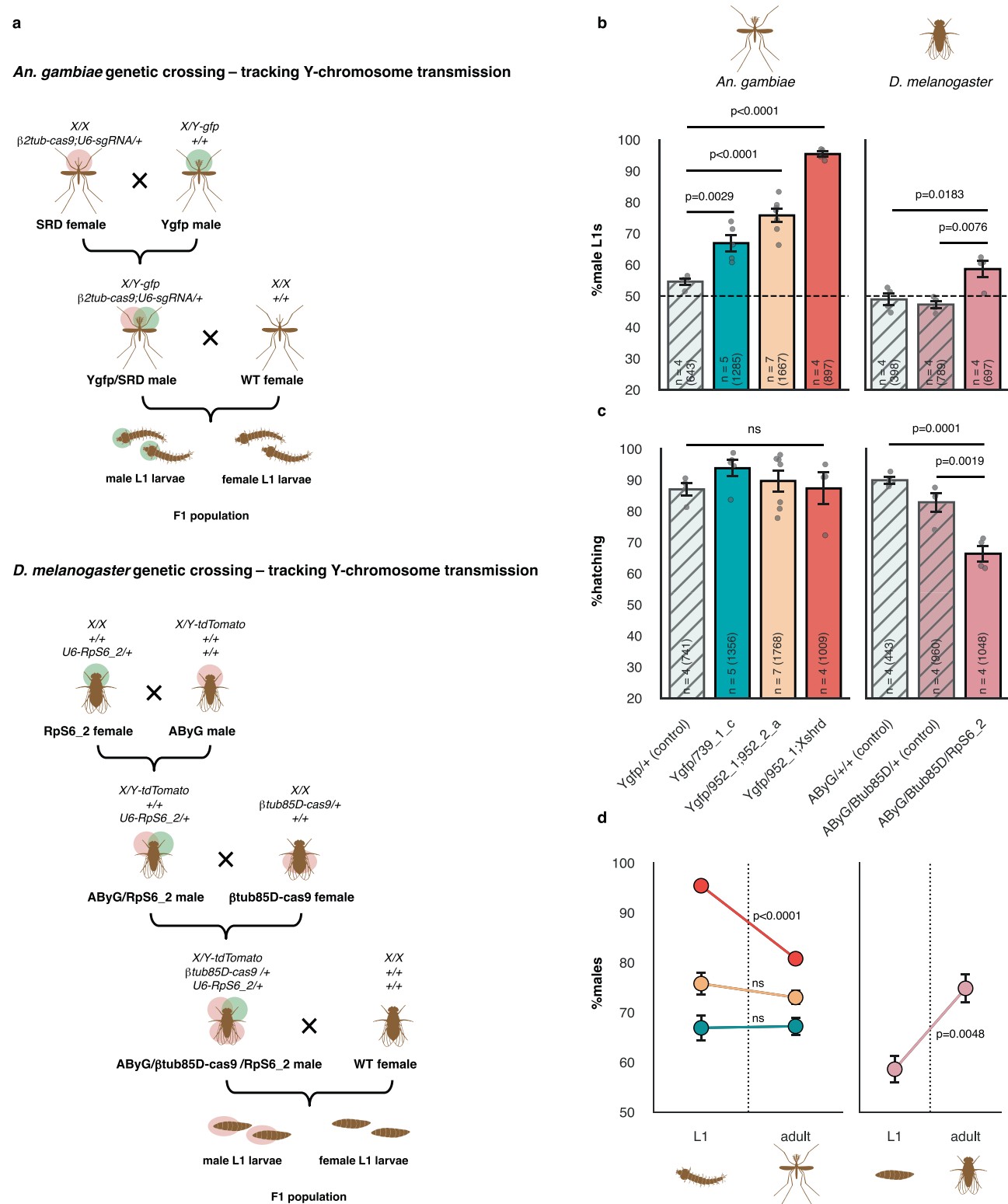

Method, $p = 0.0029$, $p < 0.0001$ and $p < 0.0001$ for 739_1_c, 952_1;952_2_a, and 952_1;Xshrd, respectively) (Fig. 4b). Hatching rates, as well as the number of eggs laid per female, were normal and did not differ from the control (Fig. 4c and Figure S5a). Sex ratio distortion levels observed in L1 larvae in both 739_1_c (66.9%) and 952_1;952_2_a (75.8%) were not statistically different than frequencies of adult males completing development in our previous experiment (Fig. 2e), indicating there is no substantial additional female-specific mortality

occurring during post-embryonic stages of development (Fig. 4d). This was confirmed by directly comparing pupation and emergence rates of male and female F1s revealing no significant differences in relative survival (Figure S5b). Interestingly, in the combined X-poisoning/X-shredding strain, the percentage of male L1s (95.4%) was higher than at the adult stage (t-test, $p < 0.0001$) (Fig. 4d), corresponding to a small but significant reduction in the emergence rate of males from this strain compared to females (t-test, $p = 0.0368$) (Figure S5b) and is

**Fig. 4 | Comparison of sex distortion mechanisms between *An. gambiae* and *D. melanogaster*. a** Overview of genetic crosses to obtain males carrying both a Y-chromosome linked marker and SRD components in *An. gambiae* (top) and *D. melanogaster* (bottom). Transgenic individuals are marked by semi-transparent red or green circles, indicating fluorescent marker expression. (WT) wild-type **b** Mean percentage (+/-SEM) of male L1 larvae based on expression of a Y-chromosome linked fluorescent markers in both *An. gambiae* (left) and *D. melanogaster* (right). "n" indicates the number of biologically independent replicates used to derive statistics, with the total number of larvae screened provided in brackets. Statistical significance was derived within each species from a One-Way ANOVA, followed by comparisons between all pairs using Dunnet's method (*An. gambiae*) or Tukey-Kramer HSD (*D. melanogaster*) (α = 0.05). Individual data points represent data from each replicate. p-values are indicated within the figure. **c** Mean hatching rates

(+/-SEM) in F1 populations of active *An. gambiae* (left) and *D. melanogaster* (right) X-poisoning strains. "n" indicates the number of biologically independent replicates used to derive statistics, with total number of eggs counted provided in brackets. Statistical significance was derived within each species from a One-Way ANOVA, followed by comparisons between all pairs using Dunnet's method (*An. gambiae*) or Tukey-Kramer HSD (*D. melanogaster*) (α = 0.05). Individual data points represent data from each replicate. p-values are indicated within the figure. **d** Percentage of males at the L1 stage and at the adult stage in F1 populations of active *An. gambiae* (left) and *D. melanogaster* (right) X-poisoning strains. Data are presented as mean values +/-SEM. Statistical significance was derived within each species from two-sided t-tests, assuming equal or unequal variances (α = 0.05). p-values are indicated within the figure. Source data are provided as Source Data files.

possibly due to a decrease in the emergence rate of transgenic males in this strain (t-test, *p* = 0.0362) (Figure S5c).

To compare these results to previous observations in the *Drosophila* model, we re-created the best-performing split Cas9 and X-poisoning sgRNA fly strains (*β*tub85D-cas9 and RpS6_2)[6]. We crossed the sgRNA strain to a Y-chromosome linked tdTomato marker (AByG[19]) to create a trans-homozygous sgRNA stock with a marked Y, and males were then crossed to virgin Cas9 females. Triple-heterozygous males containing all X-poisoning components and the marked Y-chromosome were then crossed to wild-type virgin females (Fig. 4a). Unlike mosquito strains and as previously described[6], we observed significant decrease in embryo hatching rates (66.3%) compared to AbyG/*β*tub85D-cas9 (82.8%) and AbyG-only (89.9%) controls (Tukey-Kramer, *p* = 0.0019 and *p* = 0.0001, respectively) (Fig. 4c) coupled with a male bias within freshly hatched fly larvae (58.6% compared to 47.2% and 48.9%, Tukey-Kramer, *p* = 0.0076 and *p* = 0.0183) (Fig. 4b). Within AbyG/*β*tub85D-cas9/RpS6_2 crosses, the frequency of males among neonate larvae was significantly lower than male frequencies of adults (74.8%) (t-test, *p* = 0.0048) (Fig. 4d), indicating compounding female-specific mortality occurring during larval and/or pupal stages of development, in agreement with our original results[6]. Our results confirm that, contrary to our previous model (Fig. 1), targeting of X-linked genes in mosquito sperm induces meiotic drive resulting in enhanced transmission of Y-bearing gametes at the cost of X-chromosome inheritance, unlike in *Drosophila* where similar activity against the X-chromosome does not affect sex chromosome inheritance and postzygotic induction of female-lethality indeed forms the basis for the observed sex ratio imbalance.

## Exploring the compatibility of premeiotic Cas9 expression for meiotic drives targeting RP genes

When considering ultimate deployment of SRDs in the field, linkage to the Y-chromosome is an essential step, affording SRDs an evolutionary advantage by shielding them from negative selection and enabling Y-chromosome drive in the case of X-shredding, or stable post-release dynamics in the case of X-poisoning[8]. Since pre-meiotic expression of transgenes is thought to be similar between Y and autosomal constructs[19,20], we sought to explore how earlier, pre-meiotic sperm Cas9 expression would impact the mechanism of sex ratio distortion and male fitness in our strains targeting RP genes. To leverage the mosquito strains already at hand, we generated an additional transgenic strain expressing Cas9 from the germline-specific *zpg* regulatory elements[21] that are active from spermatogonia stages[22]. We did not include additional sgRNAs in this construct, which was marked with an eCFP transformation marker to distinguish it from our X-poisoning constructs, marked with DsRed. We inserted the *zpg*Cas9 transgene in the X1 attP site[23] on chromosome 2 L and then crossed *zpg*Cas9 males to females of the two active X-targeting strains (739_1_c and 952_1;952_2_a) and also one inactive strain, which did not induce sex ratio distortion in our earlier experiments (952_1;952_2_b).

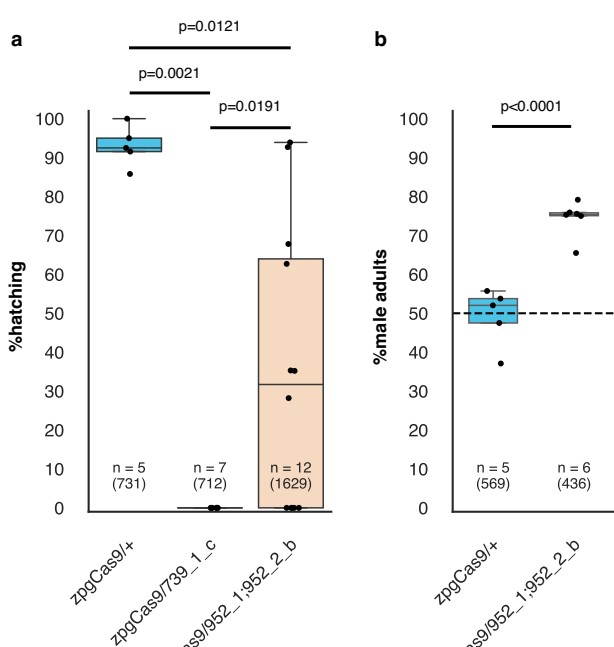

**Fig. 5 | Premeiotic expression targeting of X-chromosome targets in *An. gambiae*. a** Hatching rate and (**b**) Sex ratios of offspring from X-poisoning strains crossed with *zpg*Cas9 targeting X-linked ribosomal proteins. "n" indicates the number of biologically independent replicates used to derive statistics, with the total number of eggs and total number of adults counted in the F1 progeny, respectively, provided in brackets. Statistical analysis of hatching rates was done using the non-parametric Kruskal-Wallis Test (χ2[2] = [13.62], *p* = 0.0011), followed by comparisons of all pairs using Wilcoxon Method (α = 0.05). Statistical analysis of the percent adult F1 males was done using a one-sided t-test assuming equal variance (α = 0.05). Boxplots show median values (line), the interquartile range (IQR, box), minima & maxima (whiskers). Individual data points represent data from each replicate. p-values are indicated within the figure. Source data are provided as Source Data files.

We were not able to isolate any adult trans-heterozygotes for the cross of *zpg*Cas9 and our best SRD strain 952_1;952_2_a, as all died at the larval stage. Trans-heterozygotes expressing both markers were significantly underrepresented already at L1 larvae, compared to single marker individuals and when found were substantially smaller than larvae carrying single transgenes. By contrast, trans-heterozygotes containing the *zpg*Cas9 and either the 739_1_c or 952_1;952_2_b transgenes were viable to adulthood and occurring at the expected Mendelian frequencies. Trans-heterozygous *zpg*Cas9/739_1_c males crossed to wild-type females sired no offspring (Fig. 5a). When these males were dissected, we found abnormal, undeveloped testes that lacked mature sperm (Figure S6). On the other hand, trans-heterozygous *zpg*Cas9/

952_1;952_2_b were viable and fertile. Egg hatching rates fluctuated between replicates and were lower on average (34.6%) compared to the *zpg*Cas9/+ control (93%) (Wilcoxon Method, *p* = 0.0121) (Fig. 5a). When reared to adulthood, offspring from these males displayed a significant male bias (74.4%, t-test, *p* < 0.0001) (Fig. 5b). We found no obvious correlation between replicates with low hatching rates and those with higher male frequencies (ρ=0.3714, *p* = 0.4685), suggesting that the lower average hatching rates are not linked to the sex ratio distortion. Since on its own the 952_1;952_2_b strain had not shown sex bias in the offspring of transgenic males (Fig. 2e) and testes of this strain had low Cas9 expression according to our RNA-seq data (Figure S3a), we conclude that the activation of sex ratio distortion in trans-heterozygous males must have been driven by *zpg* driven Cas9 expression early in spermatogenesis.

## Discussion

In this study we sought to transfer the X-poisoning system to the malaria vector *Anopheles gambiae*, aiming to test approach to control mosquito populations. Following the original design from *Drosophila melanogaster*[6], we targeted three X-chromosome-linked ribosomal protein (RP) genes. We also targeted the X-chromosome with a combination of X-poisoning and X-shredding sgRNAs, predicting that sex ratio distortion levels could be boosted by eliminating females at both prezygotic and postzygotic stages, similiarly to a recent study where X-shredding was combined with a homing gene drive in the *dsx* female-specific exon[24]. To do this, we generated and then characterized multiple transgenic mosquito strains harboring both the Cas9 and single or dual sgRNAs and uncovered unexpected patterns of female elimination and sex ratio distortion. Our findings demonstrate that the X-poisoning system can be used to induce a male-biased sex ratio distortion in *An. gambiae*; however, the mechanism underlying this distortion appears to differ fundamentally from what has been described in *D. melanogaster*.

We found that independent transgenic strains demonstrated varied levels of sex ratio distortion, with three strains showing a significant male bias. Male bias among strains carrying the same construct was consistently linked with the level of Cas9 expression in male testes. Since all constructs shared the *β2-tubulin* regulatory elements to drive the expression of Cas9, differences between strains in total Cas9 expression were most likely a random outcome of the chromosomal location that the transgenic construct landed in. In the context of Cas9 expression from the Y chromosome, these results highlight the requirement of strong regulatory elements (promoters) to maximize levels of sex distortion. Using amplicon sequencing, we measured the relative activity of every sgRNA used and found that all were able to generate mutant alleles in the testes of transgenic sperm. The frequency of mutant alleles represented a minor fraction of all sequenced amplicons, but this was not surprising given that the *β2-tubulin* promoter is only activated late in spermatogenesis[25], around meiosis, and that therefore most X-chromosome alleles assayed in genomic DNA from whole testes would be derived from premeiotic sperm and further diluted somatic testicular epithelial cells. Similarly, mutant alleles were recovered only rarely from surviving adult females and not for all strains and sgRNA sites. Since our X-linked target genes should be haplolethal in nature, this was not surprising because females harboring mutation at these genes would not survive to adulthood. Indeed, when mutant alleles were present in adult females, they exclusively contained in-frame deletions. In future studies, it will be interesting to model the effect of genetic resistance of such haplolethal target genes and how they would impact a population under suppression from such a construct. It will also be interesting to attempt isolating population containing in-frame deletions in these genes to measure their actual fitness in male and female specimens.

Our initial experiments indicated a surprising absence of female-specific mortality during development in the strains displaying sex ratio distortion and targeting single RP genes. This is in contradiction

with the expected outcome of the X-poisoning mechanism, whereby mutant X-chromosomes harboring non-functional alleles of haploinsufficient genes are transmitted from transgenic males to female offspring, preventing them from completing development (Fig. 1). This unexpected observation compelled us to use a number of independent methods to explore the mechanism underlying the observed sex bias in mosquito male crosses, including single embryo genotyping and tracking of sex-specific lethality during development using Y-chromosome linked transgenic markers. We also re-generated our best-performing X-poisoning components in transgenic flies (*β*tub85D-cas9 and RpS6_2)[6] to precisely measure sex-specific lethality during development and re-testing our previous findings. Doing so validated also the use of a marked Y-chromosome enabling direct comparisons to our mosquito data. In flies, our experiments confirmed our initial conclusion that X-poisoning sex ratio distortion results from strong postzygotic lethality of female progeny during embryogenesis and later during pre-adult stages[6]. Conversely, the mosquito experiments supported a mechanism of sex bias predominantly acting as a meiotic drive, i.e. resulting in the preferential loss of X-bearing gametes and favoring the transmission of Y-bearing sperm. As such, our results are analogous to the observations for X-shredding in *An. gambiae*[2–4], *D. melanogaster*[6] and *C. capitata*[7]. Overall, our findings suggest fundamental differences exist in the pathways associated with X-chromosome breaks and repair during spermatogenesis between *An. gambiae* and *D. melanogaster*. When trying to account for these differences, it could be interesting to consider whether the lack of recombination in the male fly germline somehow protects from the loss of targeted X-gametes and enables their transmission. While many questions regarding the underlying mechanisms of preferential inheritance remain unanswered and should be further explored, our work underscores the importance of carefully accounting for the basis of sex bias when designing SRD-based genetic control systems, as these can have significant practical implications[8].

In the context of an engineered SRD, our results suggest that meiotic drives in *An. gambiae* could theoretically be built entirely agnostic of the target sequence's function or copy number on the X-chromosome. This could be explored in future studies by targeting sequences with no expected function or that are non-essential to development, for example, intronic or intergenic X-linked sequences, or an X-linked phenotypic marker. It may also be interesting to explore whether the effect we observed depends on the specific location of the target site along the X-chromosome.

Regardless of the timing of female elimination, the ultimate goal for any engineered SRD alleles would be to be physically linked to the Y-chromosome, shielding the allele from negative selection in females and ensuring transmission to all males. Previous studies have shown that expression from the *An. gambiae* Y-chromosome using the *β2-tubulin* promoter is suppressed, likely due to meiotic sex chromosome inactivation (MSCI)[26], whereas expression from promoters that are active earlier during spermatogenesis is possible from the Y, both in *An. gambiae* and *D. melanogaster*[19,20]. However, despite being conceptually the simplest way to circumvent MSCI, early expression of Cas9 may not be compatible with meiotic drive and/or male fertility. In *Drosophila*, transgenic males expressing X-poisoning sgRNAs and Cas9 driven by the *nanos* (*nos*) promoter, which is active in early germline stem cells, resulted in male sterility[6]. When combined with X-shredding, nosCas9 did not result in significant sex ratio distortion although high levels of activity could be detected. In mosquitoes, despite significant efforts focused on generating X-shredding SRDs using I-*Ppo*I or Cas9 driven by premeiotic regulatory elements like *vasa*, which are also active in early germline stem cells, a stable transgenic strain has never been isolated, likely as an outcome of induced male sterility or leaky expression in somatic tissues.

We therefore sought to test whether the meiotic drive dynamics we observed with X-poisoning sgRNAs might be compatible with early,

pre-meiotic Cas9 expression. When we crossed several X-chromosome targeting strains to an autosomal Cas9 driven by the regulatory regions of *zpg*, which drive expression in the male germline prior to *β2-tubulin*, we observed distinct outcomes: when combined with the X-poisoning strain that consistently yielded the highest levels of sex ratio distortion (952_1;952_2_a), trans-heterozygous offspring inheriting both elements were unviable with few surviving larvae displaying minute-like phenotypes, reminiscent of the *Drosophila* minute phenotype that is typical of mutants of ribosomal protein genes[27]. Since *zpg*Cas9 was introgressed from males, it is more likely that this phenotype was caused by leaky somatic Cas9 expression from one or both transgenes leading to somatic mutagenesis of *RpL37*, and not because of maternal deposition of Cas9. With the second active X-poisoning strain 739_1_c, trans-heterozygous males were sterile and contained morphologically atrophic testes without developing sperm, resembling the morphology of testes from *zpg* null or RNAi mutants[28,29]. This is similar to nosCas9/RpS6 transheterozygous males in *Drosophila*[6], suggesting that disruption of the target gene *AgRpS10* and ribosome function is intolerable at this stage of spermatogenesis. However, the addition of *zpg*Cas9 to the previously inactive strain 952_1;952_2_b, resulted in significant male-bias in the offspring. Since there was no correlation between hatching rates and sex ratio across replicates, we conclude that male-bias is likely an outcome of prezygotic meiotic drive and not by transmission of mutated X-chromosomes resulting in postzygotic lethality of female progeny.

We hypothesized that fertility of SRD males expressing premeiotic Cas9 could depend on whether their autosomal paralogs can functionally rescue the depletion of ribosomal components encoded by the X-chromosome. Our results suggest that the autosomal AGAP011501 can rescue *AgRpL37* functions, when the X-linked copy (AGAP000952) is targeted by a *zpg*-driven Cas9-SRD, but that the autosomal AGAP005469 gene cannot rescue the targeting of the X-linked *AgRpS10* (AGAP000739). This conclusion is supported by RNA-seq data measuring genome-wide expression at four stages of sperm development, namely premeiotic, early & late meiotic and postmeiotic sperm cells[30] (Figure S1b). For *AgRpL37*, the autosomal gene (AGAP011501) is more abundantly expressed throughout spermatogenesis compared to its X-linked copy (AGAP000952). The inverse was the case for *AgRpS10*, at least during early spermatogenesis, when the expression of the X-linked RP gene (AGAP000739) is around 6-fold higher than its autosomal paralog. Furthermore, at the sequence level, *AgRpL37* paralogs are more similar to each other (67% similarity, 63% identity), than the paralogs of *AgRpS10* (61% similarity, 54% identity) (Figure S2). To the best of our knowledge, this is the first demonstration of an autosomal endonuclease-based SRD expressed in early spermatogenesis, before the onset of MSCI, resulting in significant male-biased sex distortion. This finding may enable the generation of active Y-linked SRDs leveraging *zpg*Cas9 meiotic drives.

## Methods

### Ethics

All animals were handled in accordance and under the supervision of the ARO Institutional Animal Care and Use Committee approval number 2307-118-2-VOL-IL. All insect work was performed in facilities maintaining Arthropod Containment Level 2. This work received Institutional Approval and relevant authorizations from the Israel Ministry of Environmental Protection and Ministry of Agriculture (#31/2019).

### Mosquito rearing

*An. gambiae* used in this work was derived from the G3 laboratory colony of Imperial College London in November 2019. All transgenic mosquito strains were reared under standard conditions at 28 °C and 80% relative humidity at ACL-2+ containment. Larvae were provided with commercial fish food, while adult mosquitoes were given a 10% (wt/vol) glucose solution. Adult mosquitoes were allowed to mate for 4-7 days post-emergence and blood-fed on bovine blood using the

Hemotek membrane feeding system (Hemotek, Ltd). An egg-bowl was provided overnight to collect eggs 72 h post-blood-feeding. Laid eggs were placed in plastic trays containing 500 mL of demineralized water for hatching. Two days later (48 h), L1 larvae were transferred to clean trays containing 500 ml of demineralized water at a density of 250 larvae per tray and provided with fish food solution daily until pupation, according to standard protocols[31].

### Transgenic constructs

All cloning was performed using the Gibson assembly (NEBuilder HiFi DNA Assembly, New England Biolabs). PCR reactions were performed utilizing the Q5® Hot Start High-Fidelity 2X Master Mix (New England Biolabs). Plasmids were Sanger-sequenced to confirm the presence of all inserts. Primer and oligo sequences utilized for plasmid construction can be found in Table S1. To create the mosquito X-poisoning constructs, we utilized the previously described *piggyBac* vector p167[4], which includes a 3xP3-DsRed transformation marker, hCas9 driven by the *β2tubulin* promoter and 3' UTR regulatory sequence, along with a *U6*:sgRNA cassette housing a spacer cloning site bordered by *Bsa*I restriction sites on both ends. The *U6* promoter sequence used was "AnGam 2 short", as described previously by Konet et al.[15], and corresponds to the promoter of the *An. gambiae U6* gene AGAP013695. Selection of sgRNA target sites was done with CHOPCHOP[32], using the AgamP4 assembly and parameters for Cas9 knock-out. All single guide RNAs constructs were generated by annealing complementary oligos carrying unidirectional *Bsa*I compatible overhangs. The resulting plasmids, which targeted AGAP000952, AGAP000953, and AGAP000739, were named 952_2, 953_1 and 739_1 respectively. To construct the plasmids containing two distinct sgRNAs, one targeting AGAP000952 and the other of the X-poisoning and X-shredding sgRNAs, fragments containing both sgRNAs and the *U6* promoter were generated separately by gene synthesis (Genewiz), amplified and cloned into a *Bsa*I-digested p167 backbone, yielding the final vectors: 952_1;952_2, and 952_1;Xshrd, respectively. The *zpg*Cas9 construct was cloned by Gibson assembly of amplified fragments of hCas9, 3xP3 promoter, eCFP (from Addgene #117209) into an *Xho*I-digested *piggyBac* vector containing also an attB site. The intermediate plasmid was digested by *Asc*I and DNA fragments containing the *zpg* 5' and 3' regulatory regions (~1 kbp each) that were amplified from the genomic DNA were cloned. To re-build the *Drosophila* X-poisoning constructs, DNA fragments of *β*tub85D 5' and the *β*tub56D 3' regulatory region were amplified from genomic DNA of the *w*[1118] strain. The fragments were inserted into an *Xho*I and *Psy*I linearized plasmid (Addgene #112685) harboring the Cas9-T2A-eGFP expression cassette and an OpIE2-DsRed marker. The RpS6_2 sgRNA plasmid was generated following the method described in Fasulo et al.[6]. Annealed oligos were assembled into *Bbs*I-linearized plasmid (Addgene #49410) containing the *Drosophila U6:3* promoter and the 3xP3-eGFP transformation marker.

### Generation of transgenic strains

To generate mosquito transgenic strains, we performed microinjections of *An. gambiae* G3 embryos as described in Volohonsky et al.[23]. Embryos were injected with a mixture consisting of 0.4 μg/μl transformation plasmid and 0.2 μg/μl helper plasmid containing a *vasa*-driven *piggyBac* transposase. Injected larvae were screened at L1-L2 stage for transient expression of the 3xP3-DsRed marker, and positives were individually back-crossed to reciprocal wild types to obtain transgenic F1 individuals. Since *β2-tubulin* is only active in the male germline, all transgenic strains were maintained by crossing heterozygous transgenic females to wild type males. The *zpg*Cas9 strain was generated by microinjections into embryos of the X1 docking strain[23] with injection mixtures consisting of 0.33 μg/μl of the transformation plasmid and 0.08 μg/μl *vasa*-driven phiC31 integrase helper plasmid[33] (Addgene #62299). In this case, all surviving larvae were reared, and

pupae were separated into male and female cages and crossed with X1 males or females respectively. To obtain the F1 transgenic population, L1-L2 larvae were screened for eCFP and separated accordingly, after which transgenics were intercrossed to establish a homozygous population.

## Measuring sex ratio of mosquito offspring

Crosses of 50 heterozygous transgenic males from each strain and 80 wild-type females were established to measure sex ratio of adult offspring. The reciprocal female crosses (80 heterozygous transgenic females with 50 wild-type males) and a wild-type intercross were also performed as controls. To establish all crosses, mosquitoes were counted, sexed, and placed in the cross cage as pupae, where they emerged as adults. For each cross, three replicates of 250 random neonate larvae were reared to adulthood, and the total number of male and female mosquitoes was subsequently counted. To assess the stability of observed sex ratios, the experiment was repeated for at least three generations per cross. Statistical analysis was performed using a One-Way ANOVA testing the strain/cross combination's effect on the percentage of F1 adult males, followed by Tukey-Kramer HSD for comparisons between all pairs ($\alpha = 0.05$).

## Quantification of Cas9 expression by RNA-seq

Virgin 2-4 day-old transgenic males were dissected to obtain a sample of 60-80 testes, which were separated from accessory glands and other somatic tissues. As a control, a sample of 80 testes dissected from wild-type males was used. Samples were frozen in liquid nitrogen and mechanically homogenized before total RNA extraction was performed using the Quick-RNA™ Microprep Kit (Zymo Research). 200 ng from each sample were transferred to GenTegra-RNA tubes (Syntezza Bioscience, Ltd) and lyophilized with an Alfha1-4 lyophilizer (Martin Christ). mRNA libraries were prepared using poly-A capture (Syntezza Bioscience, Ltd), and samples were sequenced using the NovaSeq NGS Sequencer with paired-end reads of 150 nucleotides, generating a total of 20 million reads (Novogene Co., Ltd). Reads were mapped using HiSat2 to the *Anopheles gambiae* genome (Vectorbase, *A. gambiae* genome, version AgamP64) with the addition of the Cas9 coding sequence to quantify abundance in each sample and reads were normalized to Transcripts Per Million (TPM).

## Amplicon sequencing of targeted X-chromosomes in sperm and surviving females

To generate terminal segment samples genomic DNA was extracted from a pool of 25 posterior abdomen sections containing testes derived from 3-5 day-old adult transgenic heterozygous males, in two replicates per strain. As controls, genomic DNA was also extracted in the same manner from wild-type males. For testes samples, genomic DNA was extracted from a pool of 40 individual testes dissected from transgenic and wild-type males. For the surviving females' samples, genomic DNA was extracted from a pool of four non-transgenic virgin female progeny of transgenic males mated with wild-type females and wild-type females were used as control. PCR amplification of target regions, spanning 369-428 bp, was performed for 15 cycles across all samples using primers incorporating Illumina adapters (see Table S2 for primer sequences). Amplicons were purified using the DNA Clean & Concentrator kit (Zymo Research Corp.) and sent for additional 15 cycles of amplification and sequencing using Illumina NGS with paired-end reads and a reading depth of 100 million reads. To detect mutations at the target site, CRISPResso[34] was used on raw sequencing data. The frequency of unique alleles not matching the native sgRNA target sites in terminal segments and testes was obtained by removing alleles with a perfect match to reference sgRNA sequence, as well as alleles that appear less than 10 times in the dataset. We also removed any alleles that did not match the reference but present in wild-type samples, to obtain sequences and frequencies of CRISPR-induced mutant alleles.

## Measuring egg to adult survival

Eggs obtained from a pooled cross between 30 heterozygous transgenic males and 40 wild-type females were divided into ten replicates, with each replicate containing 50 eggs. As a control, eggs from the reciprocal female cross were included. To establish crosses, mosquitoes were counted, sexed, and placed in the cross cage as pupae, where they emerged as adults. Survival was measured daily by counting live individuals in each replicate until the emergence of adults. Survival rates were calculated for three stages: hatching (% larvae from eggs), pupation (% pupae from larvae), and adult emergence (% adults from pupae). Adults were separated by sex and counted to determine the sex ratios in each replicate. For statistical analysis we performed one-sided t-tests, assuming equal or unequal variances, comparing the survival rates of the F1 population from the male cross to those from the female cross at each stage ($\alpha = 0.05$), and comparing the percentage of F1 adults between the male and female crosses ($\alpha = 0.05$). In cases where individual replicates registered survival rates exceeding 100% at a specific stage, these were labeled as errors and excluded from the downstream statistical analysis for that specific stage (excluded: Pupa-Adult;739_1_c;male cross;replicate 5 and Pupa-Adult;739_1_c;female cross;replicate 4).

## PCR-based single embryo genotyping

To determine the sex of individual embryos, we used PCR primers 124678F2 (5'-TTTGAGCATGTGTTTAAAGG-3') and 124678R2 (5'-AGGTTTTGCCGACTACAAT-3') from Krzywinski et al.[18], binding to the AgY477 and AgX367 satellites located on the Y- and the X-chromosome, respectively. Samples were collected from 24-hour-old eggs from a pooled cross of 10 heterozygous transgenic males and 20 wild-type females. For each cross, eggs were obtained from a single cage over two sets of experiments conducted on consecutive days. Each egg was individually probed with a sterile tip to obtain the DNA sample, which was then immediately placed into a numbered PCR reaction (Platinum™ Green Hot Start PCR 2X Master Mix−Invitrogen). The sex-specific banding pattern following gel electrophoresis was evaluated for each egg to determine its sex. Samples that did not exhibit the 367 bp nonsex-specific product were labeled as unsuccessful reactions and were excluded from downstream analysis (see Table S3 for success rates). For the statistical analysis of this experiment, we employed a one-sided Fisher's Exact Test to compare the proportion of XY (male) eggs in each transgenic cross to the wild-type control ($\alpha = 0.05$).

## Tracking sex-specific survival using fluorescently marked Y-chromosome in *An. gambiae*

To obtain males carrying both the CRISPR SRD alleles and a fluorescently marked Y-chromosome, we crossed 20 transgenic females from each active SRD strain with 10 males from a strain harboring a Y-linked 3xP3-eGFP marker (Ygfp). Groups of 5 trans-heterozygous males were then mated with 5 wild-type virgin females in 4-7 replicates per cross. To establish crosses, mosquitoes were counted, sexed, and placed in the cross cage as pupae, where they emerged as adults. Eggs from each replicate were manually counted and then normalized to the number of living females within the cage, to calculate the average eggs per female number. To compare the number of eggs per female between strains, we used One-Way ANOVA. Hatching rates were calculated by manually counting hatching L1 larvae 48 hours after egg laying. To determine the proportions of males (eGFP + ) and SRD+s (DsRed + ) larvae were screened for both markers. For this statistical analysis, we used a One-Way ANOVA of the hatching rates and the percentage of male L1s between strains, followed by post-hoc comparisons of each strain against the Ygfp/+ control using Dunnet's method ($\alpha = 0.05$). To compare the percentage of males at the L1 larval stage with the percentage of males at the adult stage observed in previous experiments, we performed two-sided t-tests, assuming equal or unequal variances ($\alpha = 0.05$). To track the survival rates of male versus female offspring,

30 randomly chosen L1 larvae of each phenotype (males, females, SRD + , SRD-) were reared together in a single tray until adulthood. Survival rates were calculated at two stages: pupation (% pupae from larvae) and adult emergence (% adults from pupae). For statistical analysis, two-sided paired t-tests, assuming equal or unequal variances, were used to compare the survival rates of male individuals with that of female individuals at both pupation and adult emergence stages. Additionally, two-sided paired t-tests were conducted to compare the survival rates of SRD+ male individuals with those of SRD- male individuals at both pupation and adult emergence stages ($\alpha = 0.05$).

### Re-examining distortion and female lethality from X-poisoning in *D. melanogaster*

Fly husbandry and crosses were conducted under standard conditions at a temperature of 25 °C with a 12/12-hour day and night cycle. Fly injections were performed by Rainbow Transgenics (Camarillo, CA, USA). All genetic crosses were established using adults flies age between 1-8 hours old. The $\beta$tub85D_cas9 construct was introduced into the third chromosome using attP site VK00033 (BDSC #9750) and included an OpIE2-DsRed marker. The RpS6_2 sgRNA construct was integrated into the genome at the P{CaryP}attP40 site on the second chromosome (BDSC #25709), containing a 3xP3-eGFP marker. Transgenic strains were balanced using $w^{1118}$;Cyo;Sb/TM6B (BDSC #7197) and maintained as homozygotes. To obtain trans-heterozygous X-poisoning males, female virgins carrying the $\beta$tub85D-cas9 construct were crossed with male from the RpS6_2 sgRNA strain. At least ten trans-heterozygous RpS6_2/$\beta$tub85D_cas9 males crossed with an equal number of $w^{1118}$ females to evaluate sex ratio distortion levels by determining the percentage of males and females among adult progeny. This experiment was replicated eight times. To ascertain the developmental stage at which sex-specific mortality occurred, adult female virgins from the RpS6_2 strain were mated with males carrying a tdTomato marker linked to the Y-chromosome under the control of the 3xP3 promoter (AByG, BDSC #78567) to generate RpS6_2 males with the Y-chromosome marker. Subsequently, these males were crossed with female virgins carrying the $\beta$tub85D-cas9 construct to obtain triple-heterozygous males carrying both the X-poisoning components and the marked Y-chromosome. To assess egg hatching rates and the male ratio among L1 larvae, 20−30 triple-heterozygous males AByG/$\beta$tub85D/RpS6_2 and 20−30 $w^{1118}$ virgin females were placed in embryo collection cups with grape juice agar plates coated with yeast paste for 1-2 hours. The number of eggs and hatching larvae were subsequently counted to calculate the hatching rate. To assess the percentage of L1 males, male progeny were identified by their expression of the Y-linked 3xP3-tdTomato marker the L1 stage (figure S7). Two additional crosses involving AByG/+/+ and AByG/$\beta$tub85D_cas9/+ males crossed with $w^{1118}$ females were conducted as controls, with each cross replicated four times. Statistical analysis of hatching rates and percentage of L1 males was done using a One-Way ANOVA, followed by post-hoc comparisons of all pairs using Tukey-Kramer HSD ($\alpha = 0.05$). To compare the percentage of males at the L1 larval stage with the percentage of males at the adult stage observed in previous experiments, we performed a two-sided t-test, assuming equal variance ($\alpha = 0.05$).

### Assessing sex distortion from zpgCas9 and X-poisoning trans-heterozygotes

To obtain trans-heterozygous males expressing both *zpg*Cas9 and the X-poisoning constructs, 40 *zpg*Cas9 males were crossed to 50 X-poisoning females from each strain. When trans-heterozygous males expressing both DsRed and eCFP markers could be generated, they were crossed in groups of 40 to 50 wild-type females. As a control, a cross between male *zpg*Cas9 and wild-type females was established. For all crosses, mosquitoes were counted, sexed, and placed in the cross cage as pupae, where they emerged as adults. 48 hours after blood feeding females were separated into subgroups of 5 individuals

and were provided egg-bowls. Hatching rates were calculated by manually counting all eggs and hatching larvae from each replicate. All hatched larvae were reared to adulthood, measuring sex at the adult stage. Statistical analysis of hatching rates was done using the non-parametric Kruskal-Wallis Test, followed by comparisons of all pairs using Wilcoxon Method ($\alpha = 0.05$). Statistical analysis of percent adult F1 males was done using a one-sided t-test assuming equal variance ($\alpha = 0.05$). To test the correlation between hatching rates adult males frequencies, we used Spearman's rank correlation coefficient

### Statistics and reproducibility

Statistical tests were performed as detailed in the relevant methods sections using JMP Pro 15 software. The number of biological and/or technical replicates conducted in each experiment is described in the relevant methods sections, with each experiment being replicated a minimum of 3 times. When specific data points were excluded from downstream statistical analyses, this is clearly mentioned in the relevant methods section. No data were excluded from the Source Data files. The sample size used to derive statistics, as well as the total number of individuals tested in every experiment, is provided within each figure. All experimental crosses utilized randomly selected individuals. No statistical method was used to predetermine the sample size. The investigators were not blinded to allocation during experiments and outcome assessment.

### Reporting summary

Further information on research design is available in the Nature Portfolio Reporting Summary linked to this article.

### Data availability

The authors declare that the data supporting the findings of this study are available within the paper and its supplementary information files. Raw input data, scripts used to generate the figure panels presented in the paper, and statistical analysis files are available on figshare with the identifier https://doi.org/10.6084/m9.figshare.c.7046798[35]. Sequencing data generated by RNA-seq and amplicon sequencing analyses have been deposited in the NIH's Sequence Read Archive (SRA) with the accession code PRJNA1070274. Source data for all relevant figures are provided with the paper. Source data are provided with this paper.

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

## Acknowledgements

This work was supported by research grants from the Bill & Melinda Gates Foundation (INV-004363, PAP) and the Israel Science Foundation (ISF) (2388/19, PAP). We thank Austin Burt and Tony Nolan for helpful discussions, and the IBMC Insectarium IBiSA platform for their support in generating transgenic strains. We also thank Doron Zaada, Arad Sarig, Eli Scharlat, Neta Levine, Gleb Ens, Mai Hamburg and Moshe Elbaz for their technical assistance.

## Author contributions

Conceptualization—DAH, YA1, NW, PAP; Methodology—DAH, YA1, YA2, PAP; Resources—EK, JK, EM, PAP; Validation—DAH, YA1, PAP; Formal Analysis—DAH, YA1, PAP; Investigation—DAH, YA1, LBL, YA2, CZ, FK, ESY, RDA, PAP; Writing—Original Draft Preparation—DAH, YA1, LBL, PAP; Writing—review and editing—DAH, YA1, NW, PAP; Visualization—DAH, YA1, PAP; Supervision—PAP; Funding Acquisition—PAP.

## Competing interests

The authors declare no competing interests.
