## [Peer Review File · Nature Communications]

Reviewers' Comments:

Reviewer #1:

Remarks to the Author:

This meticulously crafted manuscript provides a comprehensive evaluation of sex ratio distorters in *Anopheles gambiae* mosquitoes. The X-poisoning versus X-shredding strategies are well explained in Fig. 1 and help the reader better understand the rest of the work. In this work, the authors attempt to translate their previous *Drosophila* expertise developing X-poisoning strategies to mosquitoes.

The authors developed a total of 9 lines to test X-poisoning, but only three of them displayed significant male bias. Interestingly, they compared lines containing only one gRNA to lines containing multiple gRNAs. No major differences were encountered in terms of the general molecular mechanisms driving male bias between these lines.

The most significant point of this work is that these lines produced male bias through X-shredding rather than X-poisoning, as the authors hypothesized during the introduction and transgenic line generation. This reviewer really appreciates the work that the authors carried out to demonstrate that their transgenic lines were driving male bias due to the generation of non-functional X-bearing sperm, as occurs in X-shredding, instead of triggering postzygotic lethality driven by X-poisoning strategies.

Finally, the authors re-evaluated their previous X-poisoning system in flies and confirmed that X-poisoning occurs differently in flies and mosquitoes. Overall, the results suggest important differences in the pathways associated with X-chromosome breaks and repair between *Anopheles gambiae* and *Drosophila melanogaster*.

I think this work will be very helpful for the CRISPR-based vector control field, which seeks to develop genome editing tools to control mosquito populations and reduce the impact of diseases such as malaria.

I have some questions and minor comments below:

Line 116: Please specify the U6 promoter.

Line 123: The authors state, "we generated a construct combining the two top-ranking sgRNAs." How do the authors know that? Did they test them before? Could you clarify this?

Lines 127-128: "We generated nine transgenic strains [...] (Figure 2B)." A quick view gives the impression that only three were generated. The authors could include a small sentence under each design. For example, under single sgRNA X-poisoning, one could add "3 independent lines." Something like this would facilitate the understanding of the figure.

Lines 147-148: It seems Cas9 levels correlate with sex ratio distortion levels. The authors could include some discussion about this observation, such as ways to increase Cas9 expression.

Fig. 3A: The Y-axis goes from 40 to 100? Could the authors clarify this?

Line 191: Could the authors explain the differences between bands in Fig.3c? While it is clear in the figure, it would be helpful to have it in the main manuscript explained to facilitate understanding.

Lines 207-211: The authors could consider adding a picture (genetic crossing) showing how the introduction of a Y-linked eGFP marker will help evaluate their CRISPR-based sex ratio distorters

further.

Fig. 4A-B: The Y-axis goes from 20 to 100? Could the authors clarify this? I am not sure why some graphs displaying hatching % go from 0 to 100 and others (like Fig. 4) not.

Consider moving Fig. S1A panel to main Fig. 2 as (b) panel. I think is helpful seeing the genes and targeted regions in the main manuscript.

Why did the authors build lines containing Cas9 and gRNAs within the same organism instead of creating a split system as they previously did in flies? Do you think this could be an issue?

How did the authors screen *Drosophila* larvae using 3xp3-GFP, which is expressed in the eyes? Did you see the GFP in the gut during larvae development as reported by others? This should be clearly explained.

A previous work described a male-biased sex-distorter gene drive in *Anopheles gambiae* (Simoni Nat Biotech 2020). While this work is not the same, the authors could include a brief discussion explaining differences and similarities (if any) between works.

Reviewer #2:

Remarks to the Author:

Review for: “Targeting mosquito X-chromosomes reveals complex transmission dynamics of sex ratio distorting gene drives”

Authors: Haber DA, Arien Y, Lamdan LB, Alcalay Y, Zecharia C, Krsticevic F, Yonah ES, Avraham RD, Krzywinska E, Krzywinski J, Marios E, Windbichler N, Papathanos PA

Reviewed by: Michael Smanski, smanski@umn.edu

Summary:

The authors present an interesting and engaging description of translating a sex-ratio biasing genetic biocontrol method from a model organism (where it worked as expected) to *Anopheles* mosquitoes (where it worked differently). In a nutshell, their ‘X-poisoning’ approach aimed to mutate haploinsufficient X-linked genes in XY male mosquitoes late in spermatogenesis. The idea is that female offspring that inherit the damaged X will die. This post-zygotic mortality was intended to offer a unique class of genetic SRB for mosquitoes in contrast to X-shredding which is known to impact the viability of X-containing sperm. A previous proof of concept in *D. melanogaster* supported their model. Surprisingly, the authors found that the X-poisoning method instead behaves like an X-shredder, with a biased transmission of Y-containing sperm during fertilization.

This paper is clearly presented, the experiments are reasonable and rigorous, and with minor exceptions the claims are well-supported by data. There are some additional experiments that could be run that would generate ‘nice to know’ information (e.g., the location of chromosomal insertion for the constructs that worked well), but I did not find any ‘need to know’ information lacking.

Overall, I recommend for publication with minor revisions. This is a well-written paper and will be of general interest to the field of genetic biocontrol. I found this paper very accessible (more so than many gene-drive related papers), which should broaden its audience. I recommend to the editors that the paper is summarized with an accompanying news article. (I would also advocate for this being published in a more field-specific Nature family journal, like Nat Biotech or Nat Genetics, but maybe that ship has already sailed). Below my signature are some suggested revisions

Best,

Michael Smanski
smanski@umn.edu

Minor revisions

1. The distinction between X-shredding and X-poisoning is clear in theory, but it is worth considering whether it is better to classify approaches at the design stage or at the empirical behavior stage. The authors seem to use the classification of the intended behavior, not the actual behavior. This is relevant because it leads to some confusing sentences in this paper. For example, the claim at the end of the introduction that “we perform direct comparisons of X-poisoning in mosquitoes and flies” seems inaccurate. The group never saw X-poisoning in

mosquitoes, and instead (serendipitously) developed a simpler and more generalizable approach for X-shredding in mosquitoes. I think this can be solved with more careful/precise language when describing what they actually built and tested.

2. Line 116 and elsewhere: Everywhere 'U6 promoter' is described, it would be best to state the specific U6 promoter that was used (unless *An. gambiae* only has a single U6 promoter).
3. Line 123; there is a reference to a ranking of sgRNAs. Please elaborate how this ranking was done (e.g., bioinformatically based on predicted off-targets, or empirically based on cutting in a transient or in vitro assay).
4. Line 196 and 325: The authors should be careful not to comment too specifically on whether the mechanism is loss of X-bearing sperm vs sperm-competition (e.g., Y-sperm out-competing the mutated X-sperm in the spermatheca). Neither mechanism can be ruled out by first principles or by data presented in this paper, as far as I can tell.
5. Line 250: Please more precise language to define 'early expression' here so that readers do not need to dig for the definition in the provided citations.
6. Line 309: Given that heritable genetic resistance was seen in the female offspring in this paper, it is worth elaborating on its impact more in the discussion. Is there a known threshold frequency of genetic resistance that would be tolerable for this approach to work in a population suppression scenario? How does this integrate with the possibility of targeting non-HI genes (as proposed in Line 339), for example targeting sequences that require more precise conservation?
7. Line 375: The presentation of the results in Figure 5 made it seem like there was a high degree of variance in hatch rate, but in the discussion, it sounds like the major source of variation was replicate number. If the latter is true (it is impossible for the reader to know for sure, because the data in Figure 5a is not reported per replicate), then it would be better not to obfuscate a known source of variance by plotting all of the data in the same column. I recommend either separating the different replicates on the X-axis, or using different shaped dots/triangles/squares to represent different replicates. Related to this data, I do not think that the ANOVA described in the methods is the appropriate statistical approach, since the % Hatching results seem to violate the prerequisite of normally distributed data. A non-parametric statistical method like the Kruskal-Wallis.
8. Figure 2: Since the n given in panel D seems to be cumulative, it is unclear what variance is being represented by SEM. Were there three replicates of n/3 each? I recommend plotting the individual measurements on top of the bar graph, similar to how datapoints were shown in Fig 5 (this may be a requirement in Nat Comm.)

Typographical

1. Line 285: recommend replacing 'our system' with 'we', since the former implies that the composite genetic system requires the concurrent targeting of three genes. Making 'systems' plural would technically be a suitable correction as well, but none of the singular systems targeted three genes.
2. Line 349: delete the 'r' in *Dr. melanogaster* (none of our flies have even passed their preliminary exams).
3. Line 371: the first clause is redundant to Line 369.
4. Line 404: do not begin the sentence with a numeral.

0. Line 410: capitalize Sanger
1. Throughout the Methods: there are several minor changes to formatting to follow traditional conventions, although to be honest I am not certain how important these conventions are any more. Examples, restriction endonucleases traditionally use italics for everything except the roman numeral (e.g., *Ascl* and *XhoI*). The fly strain ^{w¹¹¹⁸} typically uses an italics w.

Food for thought (not necessarily revision suggestions, but insight into what other questions I had when reading the manuscript)

1. It would be nice to know the chromosomal loci of the constructs in the strains that ‘worked’ versus those that did not work. For what its worth, our group (after hitting our heads against the wall for a long time trying to get inverse PCR to work) found the Splinkerette PCR method to work very well in diverse organisms and has made identifying the chromosomal location of random transgene insertion fairly easy. <https://doi.org/10.1371/journal.pone.0010168>. Alternatively, it might be worth aligning the raw data from the RNAseq experiment to the plasmid backbone. From what I have heard, there is often a small amount of contamination DNA sequenced in most RNAseq datasets, so it might be possible to learn the chromosomal loci from data that was already generated. NOTE to the editors: this comment should not be interpreted as a ‘required’ revision for acceptance for publication. It wouldn’t impact the interpretation of the presented results, but might be useful information for future Anopheles engineering work.
2. Regarding X-linked haploinsufficient genes, what is known about X-chromosome inactivation in mosquitoes? I only know the model from human biology, where at some point in development one of the X’s in females is inactivated to avoid gene dosage effects. Does this happen in mosquitoes, and if so, when? Discussing this would add interesting nuance to the mechanism of the haploinsufficiency.
3. This is probably too far afield to be worth mentioning in the current manuscript, but an interesting parallel to the X-shredder : X-poisoning methods from the world of Aquatic Invasive Species control is the work of Ron Thresher. He popularized the ‘daughterless carp’ concept in the late 1990’s early 2000’s. While the work was never made manifest in engineered carp, there was some modeling to support the importance of have 100% offspring numbers that are all phenotypic males (in his case by inhibiting a key enzyme in female hormone biosynthesis) versus 50% offspring numbers that are all male (e.g. Female lethality). The modeling work comparing female lethality vs daughterless carp is here: <https://doi.org/10.1890/07-1588.1>.
4. Figure 1: In my experience from trying to communicate the difference between something like daughterless carp and female lethal carp (see reference to Ron Thresher paper above), I have found it useful to visually represent the differences in offspring number more explicitly. I make the recommendation here solely for the author’s consideration. In Figure 1, it would be to show a population of each sperm type (e.g., copy-paste the red and blue sperm so each is represented by 3-4 individuals); show two eggs side-by-side (this is the most important recommendation: for x-poisoning two down arrows would show fertilization of one egg by the red sperm and one egg by the blue sperm), then two more down arrows showing the outcomes as you already have them (a dead red embryo next to a live blue embryo). For S-shredding, still show two eggs/embryos next to each other, but on that side of the figure there would be arrows from the blue sperm to both eggs, and there will be two blue embryos at the bottom. This will make the overall effect (twice as many offspring, and they are still all male) easier to grasp.

0. Figure 5: this is not incorrect, but I am curious why Figure 5 was plotted as box/whisker plots instead of bar graphs (as were used elsewhere in the paper). For consistency, it seems like sticking to bar graphs would have made sense.

0.

Targeting mosquito X-chromosomes reveals complex transmission dynamics of sex ratio distorting gene drives – A point-by-point response to the reviewers' comments

Reviewer #1:

This meticulously crafted manuscript provides a comprehensive evaluation of sex ratio distorters in *Anopheles gambiae* mosquitoes. The X-poisoning versus X-shredding strategies are well explained in Fig. 1 and help the reader better understand the rest of the work. In this work, the authors attempt to translate their previous *Drosophila* expertise developing X-poisoning strategies to mosquitoes.

The authors developed a total of 9 lines to test X-poisoning, but only three of them displayed significant male bias. Interestingly, they compared lines containing only one gRNA to lines containing multiple gRNAs. No major differences were encountered in terms of the general molecular mechanisms driving male bias between these lines.

The most significant point of this work is that these lines produced male bias through X-shredding rather than X-poisoning, as the authors hypothesized during the introduction and transgenic line generation. This reviewer really appreciates the work that the authors carried out to demonstrate that their transgenic lines were driving male bias due to the generation of non-functional X-bearing sperm, as occurs in X-shredding, instead of triggering postzygotic lethality driven by X-poisoning strategies.

Finally, the authors re-evaluated their previous X-poisoning system in flies and confirmed that X-poisoning occurs differently in flies and mosquitoes. Overall, the results suggest important differences in the pathways associated with X-chromosome breaks and repair between *Anopheles gambiae* and *Drosophila melanogaster*.

I think this work will be very helpful for the CRISPR-based vector control field, which seeks to develop genome editing tools to control mosquito populations and reduce the impact of diseases such as malaria.

Response: We thank the reviewer for their detailed review and supportive words.

I have some questions and minor comments below:

- Line 116: Please specify the U6 promoter.

Response: The *U6* promoter we used was previously identified by Konet et al. (2007) where it was named “AnGam 2 short”. In the updated manuscript we now make a clearer reference to the paper when first introducing the *U6* promoter used (line 116) and provide details, including AGAP identifier, in the method section “*Transgenic constructs*”.

- Line 123: The authors state, "we generated a construct combining the two top-ranking sgRNAs." How do the authors know that? Did they test them before? Could you clarify this?

Response: We did not test sgRNAs in vitro prior to selection, but relied solely on the predicted score given the *An. gambiae* genome as a reference. This is clarified better in the updated manuscript in lines 122-124, and the parameters used for CHOPCHOP prediction are detailed in the method section “*Transgenic constructs*”.

- Lines 127-128: "We generated nine transgenic strains [...] (Figure 2B)." A quick view gives the impression that only three were generated. The authors could include a small sentence under each design. For example, under single sgRNA X-poisoning, one could add "3 independent lines." Something like this would facilitate the understanding of the figure.

Response: The original reference was to the multiple designs shown in Figure 2c (previously 2B). To help clarify this better, we also added a reference to figure 2e (previously 2D), where the number of bars from each construct (of the same color) represents the number of independent lines (from the same construct).

- Lines 147-148: It seems Cas9 levels correlate with sex ratio distortion levels. The authors could include some discussion about this observation, such as ways to increase Cas9 expression.

Response: The reviewer is right about the correlation between Cas9 expression and SRD, which we state in the paper. We added a sentence in the discussion regarding the important requirement of Cas9 expression and ways to increase that.

- Fig. 3A: The Y-axis goes from 40 to 100? Could the authors clarify this?

Response: We originally did this to focus the plot on the differences in survival, given their range. In the updated manuscript, the Y-axis in this figure now goes from 0-100 to better reflect the proportions of male vs. female adult F1s that are also presented on the same axis.

- Line 191: Could the authors explain the differences between bands in Fig.3c? While it is clear in the figure, it would be helpful to have it in the main manuscript explained to facilitate understanding.

Response: we have done this in the updated manuscript in lines 189-194.

- Lines 207-211: The authors could consider adding a picture (genetic crossing) showing how the introduction of a Y-linked eGFP marker will help evaluate their CRISPR-based sex ratio distorters further.

Response: Good idea. We have added a panel to figure 4 (now Figure 4a) and cited it in the relevant section of the results, to help guide the reader.

- Fig. 4A-B: The Y-axis goes from 20 to 100? Could the authors clarify this? I am not sure why some graphs displaying hatching % go from 0 to 100 and others (like Fig. 4) not.

Response: As stated above, we did this to focus the plot on the differences, given their range. If this is not ok with the editorial team we would be happy to change it if required.

- Consider moving Fig. S1A panel to main Fig. 2 as (b) panel. I think is helpful seeing the genes and targeted regions in the main manuscript.

Response: Done (now Figure 2b).

- Why did the authors build lines containing Cas9 and gRNAs within the same organism instead of creating a split system as they previously did in flies? Do you think this could be an issue?

Response: The reviewer is correct. It took us a while until we stopped resisting the urge to split with all its advantages vs disadvantages. Most ongoing research now uses split, but back then we were still convinced that non-split is simpler to do at least in the first stage.

- How did the authors screen *Drosophila* larvae using 3xp3-GFP, which is expressed in the eyes? Did you see the GFP in the gut during larvae development as reported by others? This should be clearly explained.

Response: We have added information regarding expression of the 3xp3 marker in *Drosophila* in the relevant methods section. We have also added another new supplementary figure 7.

- A previous work described a male-biased sex-distorter gene drive in *Anopheles gambiae* (Simoni Nat Biotech 2020). While this work is not the same, the authors could include a brief discussion explaining differences and similarities (if any) between works.

Response: Done in line 290.

Reviewer #2:

The authors present an interesting and engaging description of translating a sex-ratio biasing genetic biocontrol method from a model organism (where it worked as expected) to *Anopheles* mosquitoes (where it worked differently). In a nutshell, their ‘X-poisoning’ approach aimed to mutate haploinsufficient X-linked genes in XY male mosquitoes late in spermatogenesis. The idea is that female offspring that inherit the damaged X will die. This post-zygotic mortality was intended to offer a unique class of genetic SRB for mosquitoes in contrast to X-shredding which is known to impact the viability of X-containing sperm. A previous proof of concept in *D. melanogaster* supported their model. Surprisingly, the authors found that the X-poisoning method instead behaves like an X-shredder, with a biased transmission of Y-containing sperm during fertilization.

This paper is clearly presented, the experiments are reasonable and rigorous, and with minor exceptions the claims are well-supported by data. There are some additional experiments that could

be run that would generate ‘nice to know’ information (e.g., the location of chromosomal insertion for the constructs that worked well), but I did not find any ‘need to know’ information lacking.

Overall, I recommend for publication with minor revisions. This is a well-written paper and will be of general interest to the field of genetic biocontrol. I found this paper very accessible (more so than many gene-drive related papers), which should broaden its audience. I recommend to the editors that the paper is summarized with an accompanying news article. (I would also advocate for this being published in a more field-specific Nature family journal, like Nat Biotech or Nat Genetics, but maybe that ship has already sailed). Below my signature are some suggested revisions

Best,

Michael Smanski smanski@umn.edu

Response: We thank the reviewer for their detailed review and supportive words.

Minor revisions:

- The distinction between X-shredding and X-poisoning is clear in theory, but it is worth considering whether it is better to classify approaches at the design stage or at the empirical behavior stage. The authors seem to use the classification of the intended behavior, not the actual behavior. This is relevant because it leads to some confusing sentences in this paper. For example, the claim at the end of the introduction that “we perform direct comparisons of X- poisoning in mosquitoes and flies” seems inaccurate. The group never saw X-poisoning in mosquitoes, and instead (serendipitously) developed a simpler and more generalizable approach for X-shredding in mosquitoes. I think this can be solved with more careful/precise language when describing what they actually built and tested.

Response: We totally agree with the reviewer on the difficulty and confusion generated by calling X-poisoning the same as *Drosophila* when its outcome is different in *Anopheles*. This is something we discussed a lot internally. Two reasons compelled us to stick with the current approach: The naming of X-shredding and then X-poisoning worked when initially developed in mosquitoes (for X-shredding) and then flies (both X-shredding and X-poisoning). Prior to publishing it as X-poisoning we (N. Windbichler and Papathanos) actually called the system X-meddling (to focus on the differences between

many copies of the rDNA or Muc14 or one site on a haploinsufficient gene) without implications per say on the outcome (Fasulo et al). Prior to that Papathanos and Burt called them editors and Burt published on their dynamics as YLEs (for Y-linked editors). When the X-poisoning paper was published, a reviewer suggested that meddling isn't good and perhaps poisoning would be better suited and based on available evidence at the time we agreed. In summary, the story of X-poisoning is a long and complicated one, and importantly we all feel compelled NOT to rename or rebrand as this would result in even more confusion to the lay reader at this point. We chose to stick with calling it X-poisoning in *Anopheles* to keep the emphasis (of the field of gene drive developers) on the use of the same components and orthologous targeting of the X chromosome, rather than on the outcome, as that could also depend on other factors, such as location (autosome vs Y).

- Line 116 and elsewhere: Everywhere 'U6 promoter' is described, it would be best to state the specific U6 promoter that was used (unless *An. gambiae* only has a single U6 promoter).

Response (same as reviewer #1): The *U6* promoter we used was previously identified by Konet et al. (2007) where it was named "AnGam 2 short". In the updated manuscript we now make a clearer reference to the paper when first introducing the *U6* promoter used (line 116) and provide details, including AGAP identifier, in the method section "***Transgenic constructs***".

- Line 123; there is a reference to a ranking of sgRNAs. Please elaborate how this ranking was done (e.g., bioinformatically based on predicted off-targets, or empirically based on cutting in a transient or in vitro assay).

Response (same as reviewer #1): We did not test sgRNAs in vitro prior to selection, but relied solely on the predicted score given the *An. gambiae* genome as a reference. This is clarified better in the updated manuscript in lines 122-124, and the parameters used for CHOPCHOP prediction are detailed in the method section "***Transgenic constructs***".

- Line 196 and 325: The authors should be careful not to comment too specifically on whether the mechanism is loss of X-bearing sperm vs sperm-competition (e.g., Y-sperm out-

competing the mutated X-sperm in the spermatheca). Neither mechanism can be ruled out by first principles or by data presented in this paper, as far as we can tell.

Response: We feel that loss of X-bearing sperm does not necessarily exclude loss vs the Y for example via sperm competition.

- Line 250: Please use more precise language to define ‘early expression’ here so that readers do not need to dig for the definition in the provided citations.

Response: Done in lines 257-258.

- Line 309: Given that heritable genetic resistance was seen in the female offspring in this paper, it is worth elaborating on its impact more in the discussion. Is there a known threshold frequency of genetic resistance that would be tolerable for this approach to work in a population suppression scenario? How does this integrate with the possibility of targeting non- HI genes (as proposed in Line 339), for example targeting sequences that require more precise conservation?

Response: We have added a few sentences following the reviewer’s comment (lines 313-317).

- Line 375: The presentation of the results in Figure 5 made it seem like there was a high degree of variance in hatch rate, but in the discussion, it sounds like the major source of variation was replicate number. If the latter is true (it is impossible for the reader to know for sure, because the data in Figure 5a is not reported per replicate), then it would be better not to obfuscate a known source of variance by plotting all of the data in the same column. I recommend either separating the different replicates on the X-axis, or using different shaped dots/triangles/squares to represent different replicates. Related to this data, I do not think that the ANOVA described in the methods is the appropriate statistical approach, since the % Hatching results seem to violate the prerequisite of normally distributed data. A non-parametric statistical method like the Kruskal-Wallis.

Response: We agree with the reviewer that the ANOVA is not the right method given the abnormal distribution of data points. We have now changed the method to Kruskal-Wallis

and Wilcoxon Tests and report the updated values in the relevant section in the results, as previously. We have opted not to include the changes to the figure mentioned given the lack of correlation, to save space.

- Figure 2: Since the n given in panel D seems to be cumulative, it is unclear what variance is being represented by SEM. Were there three replicates of n/3 each? I recommend plotting the individual measurements on top of the bar graph, similar to how datapoints were shown in Fig 5 (this may be a requirement in Nat Comm.).

Response: We have now added the number of replicates, as well as plotted the individual measurements in the relevant figure legends as per Nat Comm. requirements, given that this was missing. The reviewer is right that n is the cumulative number of individual eggs.

Typographical

- Line 285: recommend replacing 'our system' with 'we', since the former implies that the composite genetic system requires the concurrent targeting of three genes. Making 'systems' plural would technically be a suitable correction as well, but none of the singular systems targeted three genes.

Response: Done.

- Line 349: delete the 'r' in Dr. melanogaster (none of our flies have even passed their preliminary exams :wink:).

Response: Done. Haha

- Line 371: the first clause is redundant to Line 369.

Response: This is now clarified in the text.

- Line 404: do not begin the sentence with a numeral.

Response: Done.

- Line 410: capitalize Sanger

Response: Done.

- Throughout the Methods: there are several minor changes to formatting to follow traditional conventions, although to be honest I am not certain how important these conventions are any more. Examples, restriction endonucleases traditionally use italics for everything except the roman numeral (e.g., *AscI* and *XhoI*). The fly strain w1118 typically uses an italics w.

Response: Done.

Food for thought (not necessarily revision suggestions, but insight into what other questions I had when reading the manuscript)

1. It would be nice to know the chromosomal loci of the constructs in the strains that ‘worked’ versus those that did not work. For what its worth, our group (after hitting our heads against the wall for a long time trying to get inverse PCR to work) found the Splinkerette PCR method to work very well in diverse organisms and has made identifying the chromosomal location of random transgene insertion fairly easy. <https://doi.org/10.1371/journal.pone.0010168>. Alternatively, it might be worth aligning the raw data from the RNAseq experiment to the plasmid backbone. From what I have heard, there is often a small amount of contamination DNA sequenced in most RNAseq datasets, so it might be possible to learn the chromosomal loci from data that was already generated. NOTE to the editors: this comment should not be interpreted as a ‘required’ revision for acceptance for publication. It wouldn’t impact the interpretation of the presented results, but might be useful information for future Anopheles engineering work.

Response: Great comments and ideas. Thank you. We have also spent many a year banging our heads against the same inverse PCR to work. We will give your ideas a go.

2. Regarding X-linked haploinsufficient genes, what is known about X-chromosome inactivation in mosquitoes? I only know the model from human biology, where at some point in development one of the X’s in females is inactivated to avoid gene dosage effects. Does this happen in mosquitoes, and if so, when? Discussing this would add interesting nuance to the mechanism of the haploinsufficiency.

Response: In *Drosophila*, and likely in *Anopheles* also, gene dosage effects are solved by an overexpression of the single X-chromosome in male somatic cells.

3. This is probably too far afield to be worth mentioning in the current manuscript, but an interesting parallel to the X-shredder : X-poisoning methods from the world of Aquatic Invasive Species control is the work of Ron Thresher. He popularized the ‘daughterless carp’ concept in the late 1990’s early 2000’s. While the work was never made manifest in engineered carp, there was some modeling to support the importance of having 100% offspring numbers that are all phenotypic males (in his case by inhibiting a key enzyme in female hormone biosynthesis) versus 50% offspring numbers that are all male (e.g. Female lethality). The modeling work comparing female lethality vs daughterless carp is here: <https://doi.org/10.1890/07-1588.1>.

Response: Indeed, Burt’s models on insect control indicate the same. Initially the X-poisoning system was called daughterless (in homage) but this was eventually changed - to Phi’s great disappointment. Gender Bender was another favorite option that Phi could not get collaborators to agree on. :)

4. Figure 1: In my experience from trying to communicate the difference between something like daughterless carp and female lethal carp (see reference to Ron Thresher paper above), I have found it useful to visually represent the differences in offspring number more explicitly. I make the recommendation here solely for the author’s consideration. In Figure 1, it would be to show a population of each sperm type (e.g., copy-paste the red and blue sperm so each is represented by 3-4 individuals); show two eggs side-by-side (this is the most important recommendation: for x-poisoning two down arrows would show fertilization of one egg by the red sperm and one egg by the blue sperm), then two more down arrows showing the outcomes as you already have them (a dead red embryo next to a live blue embryo). For S-shredding, still show two eggs/embryos next to each other, but on that side of the figure there would be arrows from the blue sperm to both eggs, and there will be two blue embryos at the bottom. This will make the overall effect (twice as many offspring, and they are still all male) easier to grasp.

Response: We have taken into consideration the reviewers feedback and made changes to the figure as suggested. WE hope the reviewer likes the updated version.

5. Figure 5: this is not incorrect, but I am curious why Figure 5 was plotted as box/whisker plots instead of bar graphs (as were used elsewhere in the paper). For consistency, it seems like sticking to bar graphs would have made sense.

Response: We chose to use boxplot to emphasize in this case also the occurrence of zero values and high variation, which the bar plot generally obfuscates.